# Decline in Running Performance in Highest-Level Soccer: Analysis of the UEFA Champions League Matches

**DOI:** 10.3390/biology11101441

**Published:** 2022-10-01

**Authors:** Toni Modric, Sime Versic, Dan Iulian Alexe, Barbara Gilic, Ilie Mihai, Patrik Drid, Nikola Radulovic, Jose M. Saavedra, Rafael Burgueño Menjibar

**Affiliations:** 1Faculty of Kinesiology, University of Split, 21000 Split, Croatia; 2Department of Physical and Occupational Therapy, Faculty of Movement, Sports and Health, Sciences, “Vasile Alecsandri” University of Bacau, 600115 Bacau, Romania; 3Faculty of Kinesiology, University of Zagreb, 10000 Zagreb, Croatia; 4Department of Physical Education and Sport, Faculty of Science, Physical Education and Informatics, University of Pitesti, 110040 Pitesti, Romania; 5Faculty of Sport and Physical Education, University of Novi Sad, 21000 Novi Sad, Serbia; 6Physical Activity, Physical Education, Sport and Health Research Centre, Sports Science Department, School of Social Sciences, Reykjavik University, IS-101 Reykjavik, Iceland; 7Faculty of Education, University of Zaragoza, 50009 Zaragoza, Spain

**Keywords:** physical performance, football, fatigue, elite players

## Abstract

**Simple Summary:**

Highest-level soccer players reduce match running performance towards the end of matches, with TD being greater reduced than HIR. As players do not reduce their MRP in the latest stages of the second half, and that influence of situational factors was not observed, it seems that the decline of match running performance in highest-level soccer may not be a consequence of fatigue or situational factors but of pacing strategies.

**Abstract:**

It is widely recognized that there is a decline in match running performance (MRP) towards the end of matches. To clarify whether it is primarily a consequence of fatigue, pacing or situational influences, this study aimed to examine MRP across 15-min match periods for players on different playing positions. Players’ MRP (*n* = 244) were examined from the UEFA Champions League matches (*n* = 20) using a semiautomatic optical tracking system. Linear mixed models for repeated measures were adjusted to analyze MRP over the six 15-min match periods while controlling the influence of situational factors. No effects of match outcome, match location, team, and opponent quality on total distance (TD) and high-intensity running (HIR) for players in all playing positions were found (F = 0.03–2.75; all *p* > 0.05). Significant differences in TD (F = 17.57–53.01; η2 = 0.39–0.52, all large effect sizes) and HIR (F = 3.67–7.64; η2 = 0.05–0.19, small to medium effect sizes) among six 15-minute match periods were found for players in all playing positions. In addition, players in all playing positions covered less TD (d = 1.41–2.15, large to very large effect sizes) and HIR (d = 0.16–0.6, trivial to medium effect sizes) in the last compared to the first 15-min match period. No differences in TD and HIR between the last two match periods in the second half were observed. This study confirmed that soccer players reduce MRP towards the end of matches, and suggest that the decline of MPR in highest-level soccer may be a consequence of pacing strategies.

## 1. Introduction

Soccer is a complex team sport characterized by high physical demands [1,2,3]. A valuable method for quantifying the physical demands of soccer is an analysis of movement patterns during match play [4,5,6]. Using the technologically-advanced motion analysis systems (i.e., optical tracking systems, global and local positioning systems) [7], this most often includes analysis of match running performance (MRP), such as the total distance covered and distances covered in various speed zones (i.e., walking, jogging, running, high-intensity running, sprinting) [8,9]. Research shows that elite soccer players can cover between 9 and 14 km during the matches, performing 5–15% of that distance in high-intensity running [10,11]. 

These performances primarily vary according to the different playing positions of the players due to their different duties during the match [12,13,14]. For instance, as central midfielders (CMs) are responsible for the connection between defense and attack, these players generally achieve the greatest total distance. Players that play by the side of the pitch (i.e., fullbacks—FBs and wide midfielders—WMs) experience the most significant amount of high-intensity running due to their frequent participation in attacking actions. On the other hand, central defenders (CDs) mostly cover the lowest total- and high-intensity distance of all outfield players as their technical roles (i.e., aerial duels, tackles, positioning, interceptions) are generally more focused on the reactions instead of on the running [15,16,17,18,19]. Apart to playing positions, MRP in soccer may vary due to the various situational factors such as match location, team quality, opposition quality, or match outcome [20,21,22]. 

One of the primary research areas regarding physical demands in soccer is analysis of decline in running performance during the matches [23]. For example, analyzing differences in MRP between the first and second halves of soccer matches, some authors revealed a decline in total distance covered, high-intensity running, and the number of sprints [24,25,26]. However, a recent study reported that players do not cover less high-intensity running and sprinting distances in the second half of the matches when game interruptions are considered [27]. Studies that examined MRP across 15-min match periods reported a significant decline in high-intensity running and acceleration efforts throughout a match [28,29]. Authors demonstrated that such a decline in players’ MRP across 15-min match periods is substantially amplified by a proven increase in game interruptions [23]. A more detailed analysis of minute-by-minute observations revealed that by eight minutes into the second half, the median distance per minute had already substantially decreased compared to the corresponding median distance in the first half of minute-by-minute observations [23,30]. 

Although it is widely recognized that there is a decline in MRP towards the end of matches [31], the actual reasons for the decline in MRP are still not fully clarified [32]. Namely, some authors suggest that the decline in MRP could be due to fatigue, as studies have reported depleted muscle glycogen stores at the end of a match [32,33,34]. Other authors revealed that the decline in MRP could be due to the players employing conscious or subconscious pacing strategies to successfully complete the match [32,35,36]. On the other hand, as empirical evidence suggests that the factors such as match location, team quality, opposition quality, or match outcome strongly affect MRP [20,21,22], some authors emphasized that the decline in MRP may be related to the various situational factors [23,32]. Evidently, more research is needed to clarify whether the decline in MRP is primarily a consequence of fatigue, pacing or situational influences [32]. 

The findings from such research may also help soccer practitioners create exercises that mirror individual parts of match play, enabling players to better adapt to the high physical demands that contemporary soccer requires [37,38]. Moreover, taking into account the specificities of playing positions in the match, which is not considered in the previous studies investigating decline in MRP so far, a detailed understanding of the most intensive parts of the matches that players in different positions experience could be provided. Finally, considering a sample composed of players that competed in the most elite and most prestigious soccer clubs’ competition—the UEFA Champions League (UCL) [39], insights into the running patterns of the world’s most elite soccer players during the different periods of the matches could be drawn for the first time. Therefore, the aim of this study was to examine MRP across 15-min match periods for UCL players in different playing positions to clarify whether it is primarily a consequence of fatigue, pacing or situational influences.

## 2. Materials and Methods

### 2.1. Participants and Study Design

The sample comprised 547 individual match observations of 378 outfield players (goalkeepers were excluded due to the specificities of position) which were members of 24 teams that competed in the group stage of the UCL in the 2020/21 season. Data were collected from 20 randomly selected matches. As suggested previously, only the MRP of players who played in whole matches were analyzed, while matches that included red cards were not observed [40]. As a result, 244 MRPs were retrieved, and used as cases for this study. Players MRP were classified by match playing position as: central defenders (CD; *n* = 79), fullbacks (FB; *n* = 65), central midfielders (CM; *n* = 55), wide midfielders (WM; *n* = 28) and forward (FW; *n* = 17). 

Players’ and teams’ identities were anonymized per the principles of the Declaration of Helsinki to ensure confidentiality. The investigation was approved by the local university ethics board (approval number: 2181-205-02-05-19-0020), while written permission for the data used was obtained from Instat Limited (Limerick, Republic of Ireland, 5 June 2021). 

### 2.2. Procedures

MRP data were collected using a semiautomatic multiple-camera system InStat Fitness (Instat Limited, Limeric, Republic of Ireland). This optical system includes 3 static cameras (i.e., 2× Full HD and 1× 4K camera) installed on the roof of the soccer stadium. The system has a sampling frequency of 25 Hz, and identifies players by their movement, shape, and colour information. The system passed the official Fédération Internationale de Football Association (FIFA) test protocol for electronic and performance tracking systems (EPTS) (authorization number: 1007382), demonstrating high levels of absolute and relative reliability [41,42]. 

The MRP variables included total distance covered (TD) (m) and high-intensity running (HIR) (≥5.5 m/s) (m). To investigate temporal patterns in TD and HIR, data were divided into six pre-defined 15-min match periods: 1′–15′, 16′–30′, 31′–45′, 46′–60′, 61′–75′, and 75′–90′. Periods of extra time at the end of the first and second halves were excluded from the analysis [23].

As suggested previously, match outcome was assessed as win or not win (i.e., loss or draw) and match location as playing at home or away, while team quality, opponent quality, and differences between them were evaluated using UEFA season club coefficients [42]. The season club coefficients are based on the results of clubs competing in the current UEFA Champions League, UEFA Europa League and UEFA Europa Conference League season. Clubs’ coefficients are determined either as the sum of all points won in the previous five years OR the association coefficient over the same period, whichever is higher. Points awarded each season are in accordance with the relevant competition regulations for that specific season [42].

### 2.3. Statistical Analysis

The normality of the distributions was checked by the Kolmogorov–Smirnov test, and the data are presented as the means ± standard deviations. Levene’s test confirmed the homoscedasticity of all variables. Linear mixed models for repeated measures were adjusted to analyze TD and HIR (i.e., dependent variables) over the six 15-min match periods within playing positions while controlling the influence of match outcome, match location, team, and opponent quality. To account for players’ and teams’ multiple observations, player and team identity were modeled as a random effect [43]. Two-level dummy variables match outcome (win/not win) and match location (home/away), and continuous variables team quality and opponent quality (evaluated using UEFA season club coefficients) were introduced in the model as fixed effects, as suggested previously [42]. The assumptions of homogeneity and normal distributions of residuals were verified for each model without revealing specific problems. The main effects’ comparisons among different 15-min match periods were summarized using the least significant difference (LSD) [44,45]. Cohen’s d was used to identify effect size (ES) differences between specific 15 min match periods, interpreted as follows: <0.2, trivial; 0.2–0.6, small; 0.6–1.2, medium; 1.2–2.0, large; and >2.0, very large [46]. Partial eta squared (η2) was used to identify effect size differences among six 15-min match periods, and interpreted as follows: >0.02, small; >0.13, medium; >0.26, large. The 95% confidence intervals were computed to assess the precision of the estimates. All analyses were performed using SPSS software (IBM, SPSS, Version 25.0).

## 3. Results

The results of linear mixed models for repeated measures, adjusted to analyze MRP across the six 15-min match periods while controlling the influence of situational factors, indicated no effect of match outcome, match location, team, and opponent quality on TD and HIR for players on all playing positions (F = 0–2.89; all *p* > 0.05) (Table 1). 

Table 2 and Figure 1 present descriptive statistics and differences in TD and HIR among six 15-min match periods. Significant differences in TD (F = 17.57–53.01, all *p* < 0.01, all large ES) and HIR (F = 3.67–7.64, all *p* < 0.01, small to moderate ES) were found for players in all playing positions. In addition, significant differences in TD (F = 176.62, *p* < 0.01, large ES) and HIR (F = 18.42, *p* < 0.01, small ES) were found when players were observed independently of position.

Table 3 presents differences in TD between specific 15-min match periods. Players on all playing positions covered significantly less TD in the 16–30′ than in the 1–15′ match period (d = 0.85–1.12, all medium ES), in the 61–75′ than in 46–60′ match period (d = 1.14–1.38, all medium to large ES), and in the 75–90′ than in the 1–15′ match period (d = 1.41–2.15, all large to very large ES). In addition, CDs, FBs, CMs and WMs covered significantly less TD in 46–60′ than in 31–45′ match period (d = 0.35–0.85, small to medium ES). 

When observed independently of position, players covered significantly less TD in the 16–30′ than in 1–15′ match period (d = 0.83, medium ES), in the 46–60′ than in the 31–45′ match period (d = 0.41, small ES), in the 61–75′ than in the 46–60′ match period (d = 1.03, medium ES), and in the 75–90′ than in the 1–15′ match period (d = 1.45, large ES). 

Table 4 presents differences in HIR between specific 15-min match periods. CDs covered significantly lower HIR in the 46–60′ than in the 31–45′ match period (d = 0.38, small ES), and in the 61–75′ than in the 46–60′ match period (d = 0.51, small ES). FBs covered significantly lower HIR in the 16–30′ than in the 1–15′ match period (d = 0.51, small ES), in the 46–60′ than in the 31–45′ match period (d = 0.42, small ES), in the 61–75′ than in 46–60′ match period (d = 0.67, medium ES), and in the 75–90′ than in the 1–15′ match period (d = 0.59, small ES). CMs and WMs covered significantly lower HIR in the 16–30′ than in the 1–15′¸ match period (d = 0.45 and 0.50, respectively; both small ES), and in the 61–75′ than in the 46–60′ match period (d = 0.53, small ES and 0.84, medium ES, respectively), with WM further being covered lower HIR in the 75–90′ than in the 1–15′ match period (d = 0.60, medium ES). FW covered significantly lower HIR only in the 31–45′ than in the 16–30′ match period (d = 0.79, medium ES). 

When observed independently of position, players covered significantly lower HIR in the 16–30′ than in the 1–15′ match period (d = 0.29, small ES), in the 46–60′ than in the 31–45′ match period (d = 0.34, small ES:), in the 61–75′ than in the 46–60′ match period (d = 0.53, small ES), and in the 75–90′ than in 1–15′ match period (d = 0.33, small ES).

## 4. Discussion

This study was the first to examine MRP across 15-min match periods for UCL players in different playing positions while controlling the influence of various situational factors. No effect of match outcome, match location, team, and opponent quality on TD and HIR for players in all playing positions was found. Players in all playing positions covered lower TD and HIR in the last compared to the first match period. As this decline did not appear in the latest stage of the halves for most of the players, it seems that the decline of MPR is not a consequence of fatigue. Taken altogether, the decline of MPR in highest-level soccer is most likely a result of pacing strategies. 

Using segmentation methods, previous research repeatedly demonstrated a decline in MRP towards the end of soccer matches in different competitions [23,24,25,26,27,28,29,30]. Supportively, results from the current study indicated that UCL players in all playing positions covered less TD and HIR in the last than in the first 15-min match period. In particular, on average, players’ TD and HIR (i.e., independently of position) were lower 14% and 15%, respectively, than in the first 15-min match period. A similar study reported a 14.3% decline in the total distance in English Premier League players between the first and last 5-min match period [47]. In another study on English Premier League players, authors revealed that players covered 17.8% less HIR in the last 15-min period compared to the first [28]. Similarly, a study investigating German Bundesliga players reported 20.8% less high-speed running distance and 27.6% less sprinting distance in the same period [23]. It can be observed that the highest-level soccer players (i.e., from UCL) experience a similar decline in TD and a lower decline in HIR compared to their peers from other elite soccer competitions. Considering that UCL is one of the most physically demanding soccer competitions where players experience a large amount of HIR [37], which is associated with a greater anaerobic threshold [48], it may be assumed that a lower decline of HIR in UCL compared to other soccer competition is a consequence of players’ better conditioning status. Irrespective of causality, it must be emphasized that direct comparison of the decline in MRP with previous research is only possible to a limited extent due to the different timeframes and different definitions of speed categories chosen for investigating MRP [23]; therefore, these conclusions should be interpreted with caution. 

It is noteworthy that differences in TD between the first and last 15-min match periods were similar for players in all playing positions, while differences in HIR between the first and last 15-min match periods were position-dependent. Namely, an in-depth analysis that considered playing position revealed that CDs covered 13%, FBs 12%, CMs 15%, WMs 15%, and FWs 17% less TD in the last than in the first 15-min match period. On the other hand, greater differences in HIR were found for FBs, WMs, and FWs (21%, 19%, and 19%, respectively) than for CDs and CMs (8% and 11%, respectively). In general, it is hard for these findings to be put into perspective with the literature, as no other study has investigated position-specific decline of MRP among 15-min periods. Nonetheless, considering that FBs, WMs, and FWs generally cover the greatest HIR distance during the matches [15,17], it is most likely that a greater decline in HIR corresponds to a greater HIR distance covered during the matches. Another notable finding can be observed comparing the first and last 15-min match period. Specifically, although differences in TD and HIR between the first and last 15-min match period were similar in percentage (please see the previous Discussion for details), smaller effect sizes were found for HIR (all small to moderate) and greater for TD (all large to very large). Such findings are fully in line with previous research, which reported smaller effect sizes in higher speed categories [23]. Authors of this research explained that smaller effect sizes were associated with HIR due to the low absolute values of HIR (i.e., compared to TD and low-intensity running) during the matches [23], which was well-described in previous research investigating MRP [10,11,49,50]. Other results from the current study could directly support such findings. Namely, analyzing differences among all match periods, greater effect sizes were also found for TD (all large effect sizes) and smaller for HIR (all small to moderate effect sizes). 

Although all previously discussed confirm previous research that there is a decline in MRP towards the end of matches [23,24,25,26,27,28,29,30], the true causality is still not fully clarified. Authors speculated that the decline in MRP may be a consequence of fatigue, pacing strategies, or situational influences [32]. The first step to clarify this issue was to analyze differences in MRP across different 15-min match periods while controlling the influence of various situational factors. As previous research demonstrated that the most influential situational factors in elite soccer are match outcome, match location, team and opponent quality [51,52,53], these factors were considered in current study. Results indicated no effect of match outcome, match location, team, and opponent quality on TD and HIR for players in all playing positions. These findings suggest that the decline of MRP in highest-level soccer was not affected by herein studied situational factors but rather by fatigue or pacing strategies. For this to be further clarified, we compared MRP across subsequent match periods aiming to identify in which phase of match declines appear. Namely, as fatigue typically occurs in the latest stages of the halves [27,31,54], declines in MRP in these phases of the match may indicate that the decline was related to fatigue. However, our results show no differences in TD and HIR between the last two match periods of the first and second half for players independently of position, meaning that most of the players maintained TD and HIR toward the end of the halves. Such findings suggest that declines in MRP in highest-level soccer were not a consequence of fatigue. Here it is noteworthy that, of all playing positions, only FWs did not maintain HIR in the latest stage of the first half. Specifically, FWs covered 29% less HIR in the last 15-min match period of the first half than in the previous one (medium effect size). However, given that herein studied FWs are the world’s best players in their position whose conditioning status is at the highest possible level [37], a decline of HIR in the latest stage of the first half almost certainly does not correspond to the depleted muscle glycogen stores, which is a typical cause of fatigue [32,33,34]. Considering previous theoretical background that decline of MRP may be a consequence of fatigue, pacing or situational influences [32], while taking into account current findings which indicated that the decline of MPR in highest-level soccer was not a consequence of situational factors and fatigue, it is most likely that decline of MPR in highest-level soccer was a consequence of pacing strategies. However, as we did not directly investigate pacing strategies, these conclusions should be confirmed in future studies which will consider teams’ behaviour during the game. 

The present investigation has some limitations that should be considered. First, this study did not analyze all matches from the group stage of the UCL. Specifically, only 20 selected matches were observed. However, this is a widespread obstacle in studies involving players who compete in elite soccer [55,56]. For methodological reasons, we included only players who played the whole match, which reduced the number of observations and may have affected MRP. In addition, some of the substituted players probably left the game because of declines in MRP. Due to the limitations of the sample, we could not consider CMs as defensive and offensive midfielders. As previous research reported that this may affect results for central midfield position [57], the findings for this position should be interpreted with caution. Furthermore, due to the limitations of the tracking system used in this study, effective playing time and match interruptions, which may have a significant effect on results [23,27], were not considered. Finally, this study did not consider all factors that may have a significant effect on decline of MRP, such as partial match status during a 15 min period or goal difference. Therefore, conclusions that situational factors do not have effect on decline in MRP should not be generalizable. Future studies should address these limitations and analyze the relationship between the decline of MRP and physical fitness to confirm drawn conclusions.

## 5. Conclusions

This study confirms that highest-level soccer players reduce MRP towards the end of matches, with TD being greater reduced than HIR. Although study shows that variations on the MRP across 15-min match periods depend on players’ playing positions, it should be emphasized that herein studied players generally did not reduce their MRP in the latest stages of the second half. Therefore, decline in MPR was not probably a consequence of fatigue. Furthermore, no effect of situational factors such as match outcome, match location, team and opponent quality on MRP for players on all playing positions shows that decline in MPR was also not a consequence of herein studied situational factors. Taken altogether, the findings from this study suggest that the decline in MPR was most likely the result of pacing strategies. These findings, although necessary to be taken with caution due to the complex and unpredictable nature of soccer, may have a great deal of practical implications as they can help coaches to better understand MRP variations according to playing positions. In addition, this study’s findings may help to create exercises that mirror individual parts of match play, enabling players to adapt better to modern soccer’s high physical demands. Finally, information on position specificities may also allow greater individualization of the players’ training programs. 

## Figures and Tables

**Figure 1 biology-11-01441-f001:**
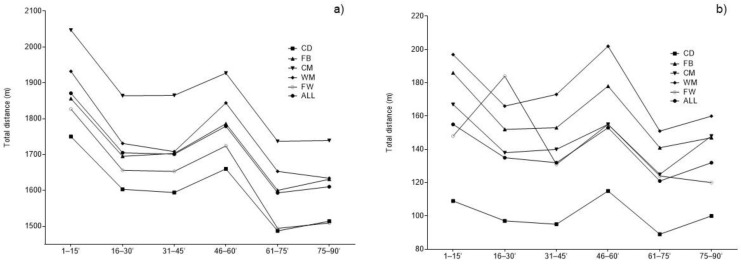
Total distance (**a**) and high-intensity running (**b**) across 15-min match periods (error bars omitted for clarity).

**Table 1 biology-11-01441-t001:** Type III tests of fixed effects (data are given as F (*p*)).

	Central Defenders	Fullbacks	Central Midfielders	Wide Midfielders	Forwards
Total distance					
Match period	53.22 (0.01)	38.8 (0.01)	42.96 (0.01)	25.94 (0.01)	17.11 (0.01)
Match outcome	0.04 (0.85)	1.1 (0.29)	0.18 (0.67)	0.22 (0.64)	0.29 (0.60)
Match location	1.39 (0.24)	2.69 (0.11)	1.02 (0.31)	1.16 (0.29)	0.06 (0.82)
Team quality	1.58 (0.22)	1.8 (0.19)	0 (0.98)	0.59 (0.45)	0.02 (0.89)
Opponent quality	0.17 (0.68)	1.61 (0.21)	2.75 (0.10)	1.68 (0.20)	0.05 (0.82)
High-intensity running					
Match period	3.73 (0.01)	7.25 (0.01)	4.75 (0.01)	3.98 (0.01)	3.84 (0.01)
Match outcome	1.77 (0.19)	2.7 (0.10)	2.81 (0.10)	0.86 (0.36)	0.03 (0.88)
Match location	1.18 (0.28)	0.05 (0.82)	0.01 (0.93)	0.95 (0.34)	0.43 (0.52)
Team quality	0.67 (0.42)	2.58 (0.12)	0.04 (0.84)	1.87 (0.19)	1.22 (0.29)
Opponent quality	2.89 (0.09)	0.07 (0.79)	0.44 (0.51)	0.63 (0.43)	0.22 (0.65)

**Table 2 biology-11-01441-t002:** Descriptive statistics and differences in total distance covered and high-intensity running among 15-min match periods (data are given as mean ± SD).

Match Period	Independently of Position	Central Defenders	Fullbacks	Central Midfielders	Wide Midfielders	Forwards
Total distance covered						
1–15′	1871 ± 193	1750 ± 141	1856 ± 169	2047 ± 152	1932 ± 183	1827 ± 168
16–30′	1705 ± 209	1603 ± 192	1695 ± 206	1864 ± 174	1731 ± 150	1656 ± 172
31–45′	1701 ± 187	1594 ± 148	1703 ± 161	1865 ± 179	1708 ± 155	1653 ± 141
46–60′	1779 ± 192	1660 ± 160	1786 ± 155	1927 ± 172	1844 ± 174	1724 ± 181
61–75′	1593 ± 168	1487 ± 142	1600 ± 135	1737 ± 138	1653 ± 130	1494 ± 152
75–90′	1610 ± 166	1514 ± 134	1631 ± 149	1739 ± 134	1634 ± 153	1509 ± 155
f (*p*)	176.62 (0.01)	53.01 (0.01)	40.66 (0.01)	42.54 (0.01)	26.04 (0.01)	17.57 (0.01)
η2	0.42	0.40	0.39	0.44	0.49	0.52
High-intensity running						
1–15′	155 ± 73	109 ± 54	186 ± 72	167 ± 71	197 ± 66	148 ± 53
16–30′	135 ± 64	97 ± 45	152 ± 60	138 ± 58	166 ± 59	184 ± 84
31–45′	132 ± 59	95 ± 50	153 ± 58	140 ± 54	173 ± 47	131 ± 44
46–60′	153 ± 66	115 ± 54	178 ± 60	155 ± 60	202 ± 73	155 ± 42
61–75′	121 ± 55	89 ± 48	141 ± 50	125 ± 52	151 ± 46	124 ± 64
75–90′	132 ± 66	100 ± 60	147 ± 60	148 ± 69	160 ± 56	120 ± 64
f (*p*)	18.42 (0.01)	3.67 (0.01)	7.64 (0.01)	4.78 (0.01)	3.99 (0.01)	3.79 (0.01)
η2	0.07	0.05	0.10	0.08	0.13	0.19

Note: f = f value; *p* = level of significance; η2 = partial eta squared effect size.

**Table 3 biology-11-01441-t003:** Differences in total distance covered between specific 15-min match periods.

Match Period	Independently of Position			Central Defenders			Fullbacks		
	MD (95%CI)	%	d	MD (95%CI)	%	d	MD (95%CI)	%	d
16–30′ vs. 1–15′	−166 (−188, −146)	−9	0.83	−147 (−185, −109)	−8	0.87	−161 (−200, −121)	−9	0.85
31–45′ vs. 16–30′	−4 (−28, 21)	0	0.02	−9 (−52, 34)	−1	0.05	8 (−42, 58)	0	0.04
46–60′ vs. 31–45′	78 (56, 100)	5	0.41	66 (33, 98)	4	0.43	83 (−129, −37)	5	0.53
61–75′ vs. 46–60′	−186 (−205, −168)	−10	1.03	−173 (−203, −143)	−10	1.14	−186 (−223, −149)	−10	1.28
75–90′ vs. 61–75′	17 (−35, 1)	1	0.10	27 (−7, 62)	2	0.20	31 (−3, 64)	2	0.22
75–90′ vs. 1–15′	−261 (−282, −242)	−14	1.45	−236 (−270, −201)	−13	1.72	−225 (−266, −183)	−12	1.41
**Match Period**	**Central Midfielders**			**Wide Midfielders**			**Forwards**		
	**MD (95%CI)**	**%**	**d**	**MD (95%CI)**	**%**	**d**	**MD (95%CI)**	**%**	**d**
16–30′ vs. 1–15′	−183 (−234, −133)	−9	1.12	−201 (−249, −153)	−10	1.2	−171 (−261, −82)	−9	1.01
31–45′ vs. 16–30′	1 (−59, 60)	0	0.01	−23 (−86, 40)	−1	0.15	−3 (−104, 98)	0	0.02
46–60′ vs. 31–45′	62 (10, 114)	3	0.35	136 (63, 208)	8	0.83	71 (−24, 166)	4	0.44
61–75′ vs. 46–60′	−190 (−235, −146)	−10	1.22	−191 (−256, −127)	−10	1.24	−230 (−304, −155)	−13	1.38
75–90′ vs. 61–75′	2 (−37, 42)	0	0.01	−19 (−73, 36)	−1	0.13	15 (−51, 80)	1	0.1
75–90′ vs. 1–15′	−308 (−351, −264)	−15	2.15	−298 (−351, −244)	−15	1.77	−318 (−383, −254)	−17	1.97

Note: MD = mean difference; 95%CI = confidence period; % = percentage difference; d = Cohen’s d effect size.

**Table 4 biology-11-01441-t004:** Differences in high-intensity running between specific 15-min match periods.

Match Period	Independently of Position			Central Defenders			Fullbacks		
	MD (95%CI)	%	d	MD (95%CI)	%	d	MD (95%CI)	%	d
16–30′ vs. 1–15′	−20 (−30, −12)	−13	0.29	−12 (−25, 2)	−11	0.24	−34 (−53, −18)	−18	0.51
31–45′ vs. 16–30′	−3 (−12, 7)	−2	0.05	−2 (−16, 11)	−2	0.04	1 (−18, 21)	1	0.02
46–60′ vs. 31–45′	21 (13, 30)	16	0.34	20 (8, 32)	21	0.38	25 (5, 44)	16	0.42
61–75′ vs. 46–60′	−32 (−24, −42)	−21	0.53	−26 (−41, −10)	−23	0.51	−37 (−56, −18)	−21	0.67
75–90′ vs. 61–75′	11 (4, 19)	9	0.18	11 (−3, 25)	12	0.20	6 (−10, 22)	4	0.11
75–90′ vs. 1–15′	−23 (−33, −14)	−15	0.33	−9 (−24, 6)	−8	0.16	−39 (−58, −20)	−21	0.59
**Match Period**	**Central Midfielders**			**Wide Midfielders**			**Forwards**		
	**MD (95%CI)**	**%**	**d**	**MD (95%CI)**	**%**	**d**	**MD (95%CI)**	**%**	**d**
16–30′ vs. 1–15′	−29 (−48, −9)	−17	0.45	−31 (−58, −5)	−16	0.50	36 (−3, 76)	24	0.51
31–45′ vs. 16–30′	2 (−18, 22)	1	0.04	7 (−21, 36)	4	0.13	−53 (−98, −9)	−29	0.79
46–60′ vs. 31–45′	15 (−3, 33)	11	0.26	29 (−1, 58)	17	0.47	24 (−1, 47)	18	0.56
61–75′ vs. 46–60′	−30 (−44, −15)	−19	0.53	−51 (−84, −19)	−25	0.84	−31 (−61, 1)	−20	0.57
75–90′ vs. 61–75′	23 (6, 40)	18	0.38	9 (−17, 35)	6	0.18	−4 (−43, 34)	−3	0.06
75–90′ vs. 1–15′	−19 (−41, 4)	−11	0.27	−37 (−70, −5)	−19	0.60	−28 (−68, 12)	−19	0.48

Note: MD = mean difference; 95%CI = confidence period; % = percentage difference; d = Cohen’s d effect size.

## Data Availability

Data will be provided to all interested parties upon reasonable request.

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
