# Peer review of "Decline in Running Performance in Highest-Level Soccer: Analysis of the UEFA Champions League Matches"

_biology, 2022, doi:10.3390/biology11101441_

Round 1

Reviewer 1 Report

Congrats for the paper.

One about the positions.

You only considerer midfielders, it is very different the role and the physical demands of the holding midfielders and attacking midfielders. Why did you switch in these two categories?

I just some question about that. You include contextual factors like the score, but you only used the final status. It is not the same if you win +1 or +3 or -2 than -1. You must considered in your variables these type of the status.

And other thing why did you have not included the partial score in each 15 min periods?

I liked too match the conclusions and the limitations.

Author Response

Congrats for the paper.

RESPONSE: Thank you very much for recognizing potential of our work. 

One about the positions.

You only considerer midfielders, it is very different the role and the physical demands of the holding midfielders and attacking midfielders. Why did you switch in these two categories?

RESPONSE: Thank you very much for this comment. We agree with you and we need to say that we were fully aware of this limitation before conducting the study. The main reason why we did not split central midfielders in defensive and offensive midfielders is actually our limited sample, which included only 55 observations from players who played in the midfield. We were opinion that splitting these 55 observations furtherly may influence the possibility of finding statistically significant differences within “new” playing positions. According to this issue, we decided to follow methodologies from many previous works which consider players who play in the midfield as “only” central midfielders. Please see some the following research:

  1. Di Salvo V, Baron R, González-Haro C, Gormasz C, Pigozzi F, and Bachl N. Sprinting analysis of elite soccer players during European Champions League and UEFA Cup matches. J Sports Sci 28: 1489-1494, 2010.
  2. Lorenzo-Martinez M, Kalén A, Rey E, López-Del Campo R, Resta R, and Lago-Peñas C. Do elite soccer players cover less distance when their team spent more time in possession of the ball? Science and Medicine in Football: 1-7, 2021.
  3. Mallo J, Mena E, Nevado F, and Paredes V. Physical demands of top-class soccer friendly matches in relation to a playing position using global positioning system technology. Journal of human kinetics 47: 179-188, 2015.
  4. Modric T, Versic S, Winter C, Coll I, Chmura P, Andrzejewski M, Konefał M, and Sekulic D. The effect of team formation on match running performance in UEFA Champions League matches: implications for position-specific conditioning. Science and Medicine in Football: 1-8, 2022.

According to all these studies, our approach seems valid. However, according to your comment, we decided to emphasize this limitation by adding following text in the Discussion: “Due to the limitations of the sample, we could not consider CMs as defensive and offensive midfielders. As previous research reported that this may affect results for central midfield position [58], the findings for this position should be interpreted with caution”. (please see Discussion, last paragraph)

I just some question about that. You include contextual factors like the score, but you only used the final status. It is not the same if you win +1 or +3 or -2 than -1. You must considered in your variables these type of the status.

RESPONSE: Thank you for these comments. We need to say that we did not include goal difference at the end of the match because our initial idea was to consider only contextual factors which are in previous research found to largely influence match running performance in soccer. Specifically, according to the literature, the most influential factors that affect MRP in soccer are match outcome, match location, team quality and opponent quality. Please see following research: 10.2478/v10078-012-0082-9; 10.1080/17461390903273994.

Also, we must say that this was already emphasized in Discussion section. Text reads: “As previous research demonstrated that the most influential situational factors in elite soccer are match outcome, match location, team and opponent quality, these factors were considered in current study” (please Discussion, 4th paragraph).

Finally, although we agree with your statement that “it is not the same if you win +1 or +3 or -2 than -1”, we must say that, to best of our knowledge, there is a lack of theoretical background which explain relationship between goal difference and match running performance. For example, Bradley et al. (2013) in their study reported that “high-intensity running and sprinting were similar in matches that were competitive, heavily won or lost” (please see 10.1080/02640414.2013.796062; table 2). On the other hand, as match outcome, assessed by final status of the match, was demonstrated to largely influence match running performance, our choice to include as fixed effect seems very logical and reasonable.

And other thing why did you have not included the partial score in each 15 min periods?

RESPONSE: For the similar reasons provided before, we did not include partial score in each 15 min period. As we said, our initial idea was to consider only contextual factors which are in previous research found to largely influence match running performance in soccer. Yes, soccer is complex and multifactorial sport, and in general soccer match performance (physical, technical, tactical, etc) largely dependent of plenty of variables except these that we included in our study. For example, playing style, team formations, level of competition, league ranking or even environmental factors (temperature, humidity, etc) can affect soccer performance. Also, partial status may also affect soccer performance. We are aware that we did not control all factors that may affect result, and that our study design is not “perfect”. However, we are also of opinion that is impossible to consider all factors that affect match performance in one article. Even is questionable whether is possible at all. This is actually highly recognized by various authors who are experts in field of running performance in socccer. For example, Linke et al. (2018) and Carling et al. (2013) agree that “physical performance in football is influenced by a great number of factors, all of which can hardly be considered collectively”. In addition, Linke et al. (2018) and Paul et al. (2015) agree that that “no single study would be able to comprehensively measure and control for all extraneous influences”. Please see:

  1. Carling C. Interpreting physical performance in professional soccer match-play: should we be more pragmatic in our approach? Sports Medicine 43: 655-663, 2013.
  2. Linke D, Link D, Weber H, and Lames M. Decline in match running performance in football is affected by an increase in game interruptions. Journal of sports science & medicine 17: 662, 2018.
  3. Paul DJ, Bradley PS, and Nassis GP. Factors affecting match running performance of elite soccer players: shedding some light on the complexity. International journal of sports physiology and performance 10: 516-519, 2015.

As we already decided to include match outcome as fixed effect, and provided valid reasons for this decision, including additional fixed effect which represent “match status” in model that already includes other “match status” factor (i.e., match outcome) seems theoretically inappropriate. To check this, we analysed the importance of adding new fixed effect by whether its inclusion demonstrated statistically significant improvements by likelihood ratio test method. Also, the Akaike information criterion (AIC) and degrees of freedom for each model (first without fixed effect “partial score”; second with fixed effect “partial score”) were visually compared (i.e., lower AIC represented a better model fit). As we found that adding new fixed effect “partial score” did not significantly improve model fit, we are of opinion that our current study design is reasonable. However, considering your concerns, we decided to emphasize this as study limitations. Text reads: “Finally, this study did not consider all factors that may have a significant effect on de-cline of MRP, such as partial match status during 15min period or goal difference. Therefore, conclusions that contextual factors do not have effect on decline in MRP should not be generalizable.” (please see the last paragraph of Discussion)

I liked too match the conclusions and the limitations.

RESPONSE: Thank you very much.

Reviewer 2 Report

First of all, I congratulate to authors for well-written and well-structured manuscript. I must also say that this study offers a very interesting and useful finding for the coaches, while at same time clarify unsolved issues regarding decline of running performance in soccer. However, I suggest some major corrections to make the manuscript even better.

TITLE

I suggest to change “football” to “soccer” because you used “soccer” through the whole manuscript.

ABSTRACT

I encourage you to add qualitative descriptors for the standardized mean differences (small, moderate, etc) through the Abstract.

INTRODUCTION

Line 59: Although the introduction is well-written, I suggest to exclude few sentences about fatigue development in soccer (start of second paragraph) and keep focus re heon decline in running performance in soccer. Also, please highlight very clearly what was the novelty of this study.

Line 49: Running performance during the game vary also according to the other factors, not just playing positions. I suggest to mention it shortly.

Line 77: References needed.

METHODS

Line 101-103: I'm interesting why did you analyze just 24 teams and 20 matches? UCL group stage consisted of 8 group with 4 teams, so altogether 32 teams. You should clarify this through the text.

Line 129: Can you provide a justification as to why match outcome was not categorized as win, draw and loss as opposed to won or not won?

Line 140: In table 2 you present “partial eta squared effect size” and did not mention it part Statistical analysis.

RESULTS

In general, this section is nicely presented, just I would encourage you to add qualitative descriptors for the standardized mean differences (e.g., small, moderate, etc) through the text.

DISCUSSION

Although well-structured and nice for read, I have concerns on your main conclusion. You stated “No effect of match outcome, match location, team, and opponent quality on TD and HIR for players in all playing positions was found. Players in all playing positions covered lower TD and HIR in the last compared to the first match period. As this decline did not appear in the latest stage of the halves for most of the players, it seems that the decline of MPR in highest-level soccer is not a consequence of fatigue but pacing strategies”. You actually directly demonstrated that decline is not a consequence of fatigue or situational factors by analysis. However, as you did not investigate pacing strategies, concluding that decline in running performance is result of pacing strategies may be considered as speculation. Yes, I agree that is logical, considering previous research, that if it is not consequence of fatigue or situational factors, then it is probably consequence of pacing strategies. I suggest to emphasize this issue through the text. This is actually nicely and simply addressed in Conclusion where you stated “Considering no effect of situational factors such as match outcome, match location, team and opponent quality on TD and HIR for players on all playing positions, it seems that decline of MPR in highest-level soccer may not be consequence of fatigue or situational factors, but of pacing strategies”. I know that this is small change, maybe just in wording, but on the other hand, I think it is important to be presented.

A major problem is related to the biological component, why was not analyzed this variable or talked about this one in this paper????

Line 315: ''Author Contributions: *** Will be added later ***???????????

Acknowledgments: *** Will be added later ***????????

Author Response

First of all, I congratulate to authors for well-written and well-structured manuscript. I must also say that this study offers a very interesting and useful finding for the coaches, while at same time clarify unsolved issues regarding decline of running performance in soccer. However, I suggest some major corrections to make the manuscript even better.

RESPONSE: Thank you very much for recognizing potential of our work. 

TITLE

I suggest to change “football” to “soccer” because you used “soccer” through the whole manuscript.

RESPONSE: Amended accordingly, thank you. Title now reads “Decline in running performance in highest-level soccer; analysis of the UEFA Champions League matches”

ABSTRACT

I encourage you to add qualitative descriptors for the standardized mean differences (small, moderate, etc) through the Abstract.

RESPONSE: Thank you for this suggestion. Qualitative descriptors for the standardized mean differences are now added. Text reads: “Significant differences in TD (F=17.57–53.01; η2=0.39-0.52, all large effect sizes) and HIR (F=3.67–7.64; η2=0.05-0.19, small to medium effect sizes) among six 15-minute match periods were found for players in all playing positions. Also, players in all playing positions covered less TD (d=1.41–2.15, large to very large effect sizes) and HIR (d=0.16–0.6, trivial to medium effect sizes) in the last compared to the first 15-min match period.”

INTRODUCTION

Line 59: Although the introduction is well-written, I suggest to exclude few sentences about fatigue development in soccer (start of second paragraph) and keep focus on decline in running performance in soccer.

RESPONSE: Thank you for this suggestion, we amended this part accordingly. Text now focuses on decline in running performance in soccer, and reads: “One of the primary research areas regarding physical demands in soccer is analysis of decline in running performance during the matches [20].”

Also, please highlight very clearly what was the novelty of this study.

RESPONSE: The biggest novelties of this study are (i) analysis of decline in MRP within each of soccer-specific playing position which is not considered in the previous studies investigating decline in MRP, (ii) use of sample consisted of the most elite soccer players in the world. Considering your comment, we additionally emphasized this issue in Introduction section. Text now reads “Moreover, taking into account the specificities of playing positions in the match, which is not considered in the previous studies investigating decline in MRP so far, a detailed understanding of the most intensive parts of the matches that players in different positions experience could be provided. Finally, considering a sample composed of play-ers that competed in the most elite and most prestigious soccer clubs’ competition – the UEFA Champions League (UCL) [39], insights into the running patterns of the world’s most elite soccer players during the different periods of the matches could be for the first time drawn.” (please see last paragraph of Introduction)

Line 49: Running performance during the game vary also according to the other factors, not just playing positions. I suggest to mention it shortly.

RESPONSE: We fully agree, thank you for this comment. We added short sentence here to mention it, as you suggested. Text reads “Apart to playing positions, MRP in soccer may vary due to the various situational factors such as match location, team quality, opposition quality, or match outcome.” (please see second paragraph of Introduction)

Line 77: References needed.

RESPONSE: Reference added, thank you.

METHODS

Line 101-103: I'm interesting why did you analyze just 24 teams and 20 matches? UCL group stage consisted of 8 group with 4 teams, so altogether 32 teams. You should clarify this through the text.

RESPONSE: Thank you for this suggestion. The main reason for this is availability of data. Specifically, InStat does not record all matches in UCL. It is actually private company that record matches only on team request (and if pay it, of course). For example, some teams use their own GPS system, while other teams have own LPS systems installed on stadiums. Such teams do not need additional information on running performance. As not all teams order analysis from InStat, availability of data is limited. This is the main reason why all matches from all groups were not analysed. Apart to this, we really believe that our multi-team sample is very impressive. However, according to your comment, we clarified that matches were randomly selected. Text reads: “Data were collected from 20 randomly selected matches”.

In the end, although we were aware that this may be limitation, we must say that using relatively small sample is a very common obstacle in studies involving players which compete at the highest level of soccer (10.1080/02640414.2011.561868; 10.3390/app11188765; 10.1080/24733938.2022.2123952). At any rate, we emphasized in Limitations section at the end of discussion. Please see: “The present investigation has some limitations that should be considered. First, this study did not analyze all matches from the group stage of the UCL. Specifically, only 20 selected matches were observed. However, this is a widespread obstacle in studies involving players who compete in elite soccer [55,56]”

Line 129: Can you provide a justification as to why match outcome was not categorized as win, draw and loss as opposed to won or not won?

RESPONSE: Thank you very much for this comment. The reason why we included match outcome as two-level categorical predictor is simple. Ultimate goal of playing soccer is winning. Yes, we agree that sometimes even draw can be “positive” results, but in general drawing and losing are “unfavourable” status. With including two-level categorical predictor, it could be very easy to see how MRP change when status of the matches changes from “unfavourable” to winning and vice versa. This was, in our opinion, very interesting.

However, another tnteresting thing is that we did not find significant effect of match outcome in this study and therefore we did not provide estimates of fixed effects. But, in case that we found significant effect of match outcome, including three-level categorical predictor (win, draw, loss) would most likely require discussion between drawing and losing. And in our opinion, this is not very interesting.

Since this approach (including match outcome as two-level categorical predictor) was applied in other studies which assess situational factors, we believe that it is appropriate and relevant (please see:  10.1371/journal.pone.0247771; 10.5114/biolsport.2023.116453).

Finally, we emphasized in Methods that this approach was suggested previously. Please see following text: “As suggested previously, match outcome was assessed as win or not win (i.e., loss or draw) and match location as playing at home or away, while team quality, opponent quality, and differences between them were evaluated using UEFA season club coeffi-cients [42].” (see Procedures, 3rd paragraph)

Line 140: In table 2 you present “partial eta squared effect size” and did not mention it part Statistical analysis.

RESPONSE: Indeed, thank you for noticing this. We amended this according to your comment. Now section Statistical analysis include information on partial eta squared effect size. Text reads: “Partial eta squared (η2) was used to identify effect size differences among six 15min match periods, and interpreted as follows: >0.02, small; >0.13, medium; >0.26, large” (please Methods, Statistical analysis).

RESULTS

In general, this section is nicely presented, just I would encourage you to add qualitative descriptors for the standardized mean differences (e.g., small, moderate, etc) through the text.

RESPONSE: Thank you for this suggestion. Qualitative descriptors for the standardized mean differences through the whole Results section are now added. Text reads: “Table 3 presents differences in TD between specific 15-min match periods. Players on all playing positions covered significantly less TD in the 16–30’ than in the 1–15’ match period (d=0.85–1.12, all medium ES), in the 61–75’ than in 46–60’ match period (d=1.14–1.38, all medium to large ES), and in the 75–90’ than in the 1–15’ match period (d=1.41-2.15, all large to very large ES). In addition, CDs, FBs, CMs and WMs covered significantly less TD in 46–60’ than in 31–45’ match period (d=0.35–0.85, small to me-dium ES).

When observed independently of position, players covered significantly less TD in the 16–30’ than in 1–15’ match period (d=0.83, medium ES), in the 46–60’ than in the 31–45’ match period (d=0.41, small ES), in the 61–75’ than in the 46–60’ match period (d=1.03, medium ES), and in the 75–90’ than in the 1–15’ match period (d=1.45, large ES).”

and

“Table 4 presents differences in HIR between specific 15-min match periods. CDs covered significantly lower HIR in the 46–60’ than in the 31–45’ match period (d=0.38, small ES), and in the 61–75’ than in the 46–60’ match period (d=0.51, small ES). FBs covered significantly lower HIR in the 16–30’ than in the 1–15’ match period (d=0.51, small ES), in the 46–60’ than in the 31–45’ match period (d=0.42, small ES), in the 61–75’ than in 46–60’ match period (d=0.67, medium ES), and in the 75–90’ than in the 1–15’ match period (d=0.59, small ES). CMs and WMs covered significantly lower HIR in the 16–30’ than in the 1–15’¸ match period (d=0.45 and 0.50, respectively; both small ES), and in the 61–75’ than in the 46–60’ match period (d=0.53, small ES and 0.84, medium ES, respectively), with WM further being covered lower HIR in the 75–90’ than in the 1–15’ match period (d=0.60, medium ES). FW covered significantly lower HIR only in the 31–45’ than in the 16–30’ match period (d=0.79, medium ES).

When observed independently of position, players covered significantly lower HIR in the 16–30’ than in the 1–15’ match period (d=0.29, small ES), in the 46–60’ than in the 31–45’ match period (d=0.34, small ES:), in the 61–75’ than in the 46–60’ match period (d=0.53, small ES), and in the 75–90’ than in 1–15’ match period (d=0.33, small ES).”

DISCUSSION

Although well-structured and nice for read, I have concerns on your main conclusion. You stated “No effect of match outcome, match location, team, and opponent quality on TD and HIR for players in all playing positions was found. Players in all playing positions covered lower TD and HIR in the last compared to the first match period. As this decline did not appear in the latest stage of the halves for most of the players, it seems that the decline of MPR in highest-level soccer is not a consequence of fatigue but pacing strategies”. You actually directly demonstrated that decline is not a consequence of fatigue or situational factors by analysis. However, as you did not investigate pacing strategies, concluding that decline in running performance is result of pacing strategies may be considered as speculation. Yes, I agree that is logical, considering previous research, that if it is not consequence of fatigue or situational factors, then it is probably consequence of pacing strategies. I suggest to emphasize this issue through the text. This is actually nicely and simply addressed in Conclusion where you stated “Considering no effect of situational factors such as match outcome, match location, team and opponent quality on TD and HIR for players on all playing positions, it seems that decline of MPR in highest-level soccer may not be consequence of fatigue or situational factors, but of pacing strategies”. I know that this is small change, maybe just in wording, but on the other hand, I think it is important to be presented.

RESPONSE: Thank you very much for this comment. As we fully agree with you, we amended parts of the Discussion and Conclusions to emphasize this. Please see following text:

In Discussion, 1st paragraph: “No effect of match outcome, match location, team, and opponent quality on TD and HIR for players in all playing positions was found. Players in all playing positions cov-ered lower TD and HIR in the last compared to the first match period. As this decline did not appear in the latest stage of the halves for most of the players, it seems that the decline of MPR is not a consequence of fatigue. Taking altogether, the decline of MPR in highest-level soccer is most likely result of pacing strategies.”

In Discussion, 4th paragraph: “Considering theoretical background from previous that decline of MRP may be consequence of fatigue, pacing or situational influences [32], while taking into account current findings which indicated that decline of MPR in highest-level soccer was not consequence of situational factors and fatigue, it is most likely that decline of MPR in highest-level soccer was consequence of pacing strategies. However, as we did not directly investigate pacing strategies, these conclusions should be confirmed in future studies which will consider teams’ behaviour during the game.”

In Conclusion: “This study confirms that highest-level soccer players reduce MRP towards the end of matches, with TD being greater reduced than HIR. Although study shows that vari-ations on the MRP across 15-min match periods depend on players’ playing positions, it should be emphasized that herein studied players generally did not reduce their MRP in the latest stages of the second half. Therefore, decline in MPR was not proba-bly a consequence of fatigue. Further, no effect of situational factors such as match outcome, match location, team and opponent quality on MRP for players on all playing positions shows that decline in MPR was also not be consequence of herein studied situational factors. Taking altogether, the findings from this study suggest that the de-cline in MPR was most likely result of pacing strategies.”

A major problem is related to the biological component, why was not analyzed this variable or talked about this one in this paper????

RESPONSE: Thank you for this comment. We must say that we do not agree with you that “major problem is related to the biological component”. The aim of this research was to examine MRP across 15-min match periods for players on different playing positions, in order to clarify whether is decline in MRP primarily a consequence of fatigue, pacing or situational influences. Since previous research investigating decline in MRP in soccer did not provide any relevant theoretical background that decline in MRP may be related to the “biological component”, we did not consider it. This is the reason “why this variable was not analysed”.

Line 315: ''Author Contributions: *** Will be added later ***???????????

Acknowledgments: *** Will be added later ***????????

RESPONSE: Author Contributions and acknowledgments are now added.

Round 2

Reviewer 2 Report

The authors have considerably improved the article and the paper it be potential for publication.